# Wound Detection by Simple Feedforward Neural Network

**Domagoj Marijanović** *⬤, **Emmanuel Karlo Nyarko** ⬤ and **Damir Filko**

Faculty of Electrical Engineering, Computer Science and Information Technology Osijek, 31000 Osijek, Croatia; karlo.nyarko@ferit.hr (E.K.N.); damir.filko@ferit.hr (D.F.)
* Correspondence: domagoj.marijanovic@ferit.hr

**Abstract:** Chronic wounds are a heavy burden on medical facilities, so any help in treating them is most welcome. Current research focuses on wound analysis, especially wound tissue classification, wound measurement, and wound healing prediction to assist medical personnel in wound treatment, with the main goal of reducing wound healing time. The first phase of wound analysis is wound segmentation, where the task is to extract wounds from the healthy tissue and image background. In this work, a standard feedforward neural network was developed for the purpose of wound segmentation using data from the MICCAI 2021 Foot Ulcer Segmentation (FUSeg) Challenge. It proved to be a simple yet efficient method for extracting wounds from images. The proposed algorithm is part of a compact system that analyzes chronic wounds using a robotic manipulator, RGB-D camera and 3D scanner. The feedforward neural network consists of only five fully connected layers, the first four with Rectified Linear Unit (ReLU) activation functions and the last with sigmoid activation functions. Three separate models were trained and tested using images provided as part of the challenge. The predicted images were post-processed and merged to improve the final segmentation performance.The accuracy metrics observed during model training and selection were Precision, Recall and F1 score. The experimental results of the proposed network provided a recall value of 0.77, precision value of 0.72, and an F1 score (Dice score) of 0.74.

**Keywords:** chronic wounds; wound detection; wound segmentation; feedforward neural network; robot



## 1. Introduction

Chronic wounds are defined as wounds that do not heal properly and therefore require special treatment. Such wounds may be the result of diabetes, venous ulcers, foot ulcers, burns, etc. Due to the complexity of wounds, patients must stay in medical centers for a long period of time, so cost of treating such patients can be extremely high. Wound analysis is usually done manually with rulers or by visual inspection, which depends on the expertise of a doctor or other medical personnel. With the rapid development of technology, devices such as mobile phones and high-precision cameras have motivated the research community to explore the possibilities of such devices. In order to accelerate the wound healing process and facilitate treatment, researchers are using image processing for wound analysis, among other approaches [1]. Deep-learning methods using images of wounds as input data can be used to analyze the wound and calculate the healing process based on the tissue representation and values of wound metrics. The first stage of wound analysis is segmentation, where the wound is separated from the healthy tissue and background. The accuracy of the segmentation is important because all other subsequent steps are based on the extracted wound tissue. The next steps in wound analysis include tissue classification (the three main classes are granulation, fibrin and necrosis), 3D reconstruction, wound measurement and healing prediction. Often, to achieve better results, researchers use preprocessing methods such as the conversion of color spaces, noise removal, color correction, etc., before the first stage of wound tissue separation [2].

The model proposed in this paper is developed using data from the Foot Ulcer Segmentation Challenge [3] whose goal was to perform wound detection from images photographed in medical centers. The dataset provided has more than 1000 images collected in medical centers under natural conditions.

Wound detection with a pixel-level instance segmentation model presented in this paper is part of a more complex system, which is currently being developed and which will conduct a complete analysis of the wound, i.e., wound detection, 3D reconstruction, tissue classification, and wound measurement. The system consists of a 7 DoF robot arm with an attached RGB-D camera and a high-precision 3D scanner as shown in Figure 1. The first two stages of the wound-analysis system have currently been developed: the first stage being wound detection, which is described in this paper, and the second stage being wound reconstruction, which has been developed using the newly proposed NBV algorithm that allows medical personnel to inspect the wound from any angle [4]. Wound detection as the initial phase is crucial for the overall system's precision and efficiency. The input to the wound-detection stage is an image of the patient, and the output is the same image with all detected wounds marked using bounding boxes. The largest wound detected, determined by the largest bounding box, is specifically marked and all the proceeding steps of the wound-analysis system are performed on this detected wound unless otherwise specified by the user of the system. Wound detection must be precise, but moreover, it must always detect all wounds on the image, so the most important metric used in this stage is the recall metric value since it is imperative that all wounds on the image are detected, i.e., there should not be false negatives.

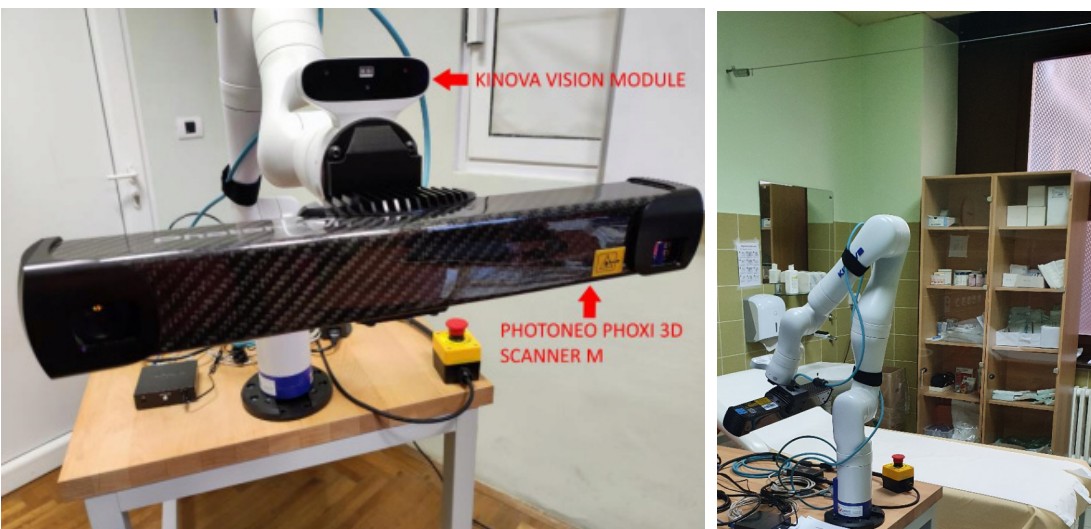

**Figure 1.** The figure shows an automated system for wound analysis consisting of a 7 DoF (Degrees of Freedom) robot arm, Kinova RGB-D camera, and a high-precision 3D scanner used for the recording of chronic wounds in the medical center.

The wound-detection algorithm in this paper is developed as a simple feedforward neural network with four fully connected hidden ReLU layers and one fully connected output layer with a sigmoid activation function. We propose that since wounds are highly irregular and can be of any shape or texture, simple feedforward neural networks should suffice for wound detection and there is no need for more complicated neural networks. Since the main idea of the paper is to the analyze the feasibility of creating a suitable wound detector using a standard feedforward neural network which is as simple as possible in terms of network structure, the ReLu activation function is used in the hidden layers. Additionally, the choice of four hidden layers is relatively arbitrary, as it is the desire to have a neural network classifier complex enough to capture the possible nonlinear relationship between the output and the corresponding input data.

A standard fixed-size overlapping sliding-window procedure is used to generate input data to the feedforward neural network classifier. The output image, obtained after the whole input image is processed using this sliding-window procedure, is a probability map. This map is the same size as the input image and indicates the probability of each pixel being a wound pixel.

Post-processing, involving thresholding, noise removal, and region filling, is then performed on this probability map to obtain a binary image of the same size as the input image, where pixels marked True indicate *wound* pixels, while False indicates *non-wound* pixels.

The main contribution of this paper is a novel method for wound detection based on a sliding-window procedure and a simple feedforward neural network classifier that can detect and segment the wound area from the image with satisfactory efficiency and speed. With only five network layers, the method shows relatively high robustness to all wound shapes, positions, and types of tissues. Due to the simple network architecture, the training time of each epoch is shorter than the time required to train deep neural networks. An effective way of post-processing of the predicted images is conducted in a significantly short time.

The rest of the paper is organized as follows. Section 2 presents related research of chronic wound segmentation, Section 3 describes the proposed method, and the experimental results of the paper are provided in Section 4. The obtained results are discussed in Section 5 and, finally, the main conclusions are presented in Section 6.

## 2. Related Research

The task of wound segmentation is to separate image pixels into two classes, *wound* and *not-wound*, i.e., to extract the wound area from surrounding healthy tissue or image background. Segmentation precision is crucial for further wound-analysis tasks such as tissue classification, 3D reconstruction, wound measurement, and wound healing evaluation. Since the wound area usually has a different color to the healthy skin, extracted color features from each pixel are decisive in the extraction of the wound. Biswas et al. [5] performed wound segmentation in two stages using a support vector machine (SVM) algorithm trained on a combination of color and texture features. In the first stage, the wound is separated into two classes, background and skin class, while in the second stage the resulting skin class from the first stage is separated into two more classes, healthy skin and wound class. Wound area segmentation was performed with an accuracy of 71.98% on ten images. The author's next work described in [6] is based on a superpixel region growing algorithm and formed a 4D probability map which achieved an accuracy of 79.2% on 30 images. Dhane et al. [7] used only the S channel from the HSV color space where they converted the image into data points, and with the aid of the Gaussian similarity function, calculated a similarity graph. K-means algorithm was used to cluster the data and, based on 105 images, an accuracy of 86.71% was achieved. In a newer paper from the same authors [8], a similar workflow was used, where the calculation of the similarity matrix was performed using Gray-based fuzzy similarity measure. Li et al. [9] developed a neural network based on modified MobileNet architecture to perform wound segmentation. A high-precision value of 94.69% was obtained on 950 images. The authors continued their work [10] on MobileNet architecture but with a different number of channels and compared it with the VGG16 deep neural network. Both networks produced admirable results, e.g., MobileNet achieved an accuracy of 98.18%. In their subsequent research [11], one convolutional layer was added as the first layer in MobileNet architecture, which consisted of a convolutional kernel enhanced with location information. The achieved precision of the adapted neural network was 95.02%.

Filko et al. [12] proposed a system based on an RGB-D sensor for the detection, segmentation and 3D reconstruction of chronic wounds. Wound detection was based on color histograms and kNN algorithm. Based on the same algorithms, the authors improved upon the system in [13], where the wound contour was extracted by the wound-segmentation procedure using geometrical and visual information of the wound surface.

Gholami et al. [14] analyzed and compared seven machine-learning algorithms on wound segmentation: region-based methods (region growing and active contour without edges), edge-based methods (edge and morphological operations, level set method preserving the distance function, livewire, and parametric active contour models or snakes) and texture-based methods (Houhou–Thiran–Bresson model). Livewire method produced results with the highest score but with the longest calculation time.

Wang et al. [15] proposed a method for wound segmentation based on a deep convolutional neural network on the MobileNetV2 architecture. The dataset used for training and testing of the network consisted of 1010 images collected at the AZH Wound and Vascular Center. The dataset was preprocessed and augmented to satisfy the neural networks necessity for a large training set. Before training, the network was pre-trained on the Pascal VOC segmentation dataset. The output of the generated model was a segmented grayscale image of the wound where each pixel denoted the likelihood of being a wound. The image was thresholded using a post-processing step and the final wound area was marked. The achieved recall value was 89.97%, precision 91.01%, and F1 score 90.47%. They also compared their model to four other models, namely VGG16, SegNet, U-Net and Mask-RCNN using the Medetec dataset, and showed their model to be superior to the other models. The same authors continued their research in [16] and performed a systematic review of 115 papers covering image-based AI in wound assessment. They concluded that with different approaches used, each one has some limitations so no method could be pointed out as a method preferable to others.

Mahbod et al. [17] proposed a segmentation method that consists of two encoder–decoder convolutional neural networks, U-Net and LinkNet. Both networks were pre-trained on the Medetec database [18] on 152 images and afterwards, the model was trained on the chronic wound dataset shared by the MICCAI 2021 Foot Ulcer Segmentation Challenge containing 1010 annotated images. The results of training are two separate models generated by each network. The test image is evaluated by both models and combined into the final result. The obtained data-based F1 score was 88.8%, which represents the current state of the art.

## 3. Method

### 3.1. Dataset

Researchers in the field of wound detection/segmentation are often limited by the number of labeled images for machine-learning algorithms and methods. Most researchers use the Medetec database [18] as one of the larger annotated bases for chronic wound analysis. The proposed algorithm of wound segmentation in this paper is conducted as a pixel-based method on the dataset provided by the MICCAI 2021 Foot Ulcer Segmentation Challenge (FUSeg) [3]. The images were taken at clinical visits and photographed under natural conditions. They vary in shape, size, closeness, and background. The dataset contains 1210 total images—1010 labeled images, and 200 non-labeled images used for final testing purposes only. Labeled images are divided into 810 for training with validation and 200 for testing. Ground truth of labeled images are provided as grayscale images, where higher values denote the probability of a pixel being wound. For the method proposed in this paper, all images were resized to $224 \times 224$. The rational for resizing was only to decrease the total amount of input data. Figure 2 shows examples of wound images with corresponding annotations from the aforementioned dataset.

### 3.2. Data Generation

As mentioned in Section 1, a sliding-window procedure is used to generate data from the input image to the feedforward neural network. Let $w_{size}$ denote the width or height of the sliding window, and $w_{step}$ the step size of the sliding window i.e., the number of pixels the window shifts during one step of the sliding procedure. The term sub-image is used to describe the image region of interest (ROI) defined by the current position of the sliding window. During the training phase, a $w_{size} \times w_{size}$ sliding window is moved with

step size, $w_{step}$, across the input image and the corresponding labeled ground-truth image simultaneously, therefore generating input sub-images and corresponding ground-truth sub-images, as shown in Figure 3. An input sub-image is marked entirely as *wound* if the total percentage of *wound* pixels in the corresponding ground-truth sub-image is greater than a given threshold, $wound\_pixels_\%$, otherwise the input sub-image is marked as *not-wound*. As a result, a given number of sub-images are generated per input image, and each input sub-image is classified as *wound* or *not-wound*. Since a standard feedforward neural network is used as a classifier, each input sub-image is transformed into a 1-dimensional input vector by simply transforming each channel of the considered color space of the sub-image into a vector and then concatenating all three channels. In this paper, only the RGB color space was considered and $w_{size} = \{5, 7, 9\}$, $w_{step} = 1$ and $wound\_pixels_\% = \{25\%, 50\%\}$. Training data were generated using the aforementioned procedure on all 810 labeled training images from the MICCAI 2021 Foot Ulcer Segmentation Challenge dataset.

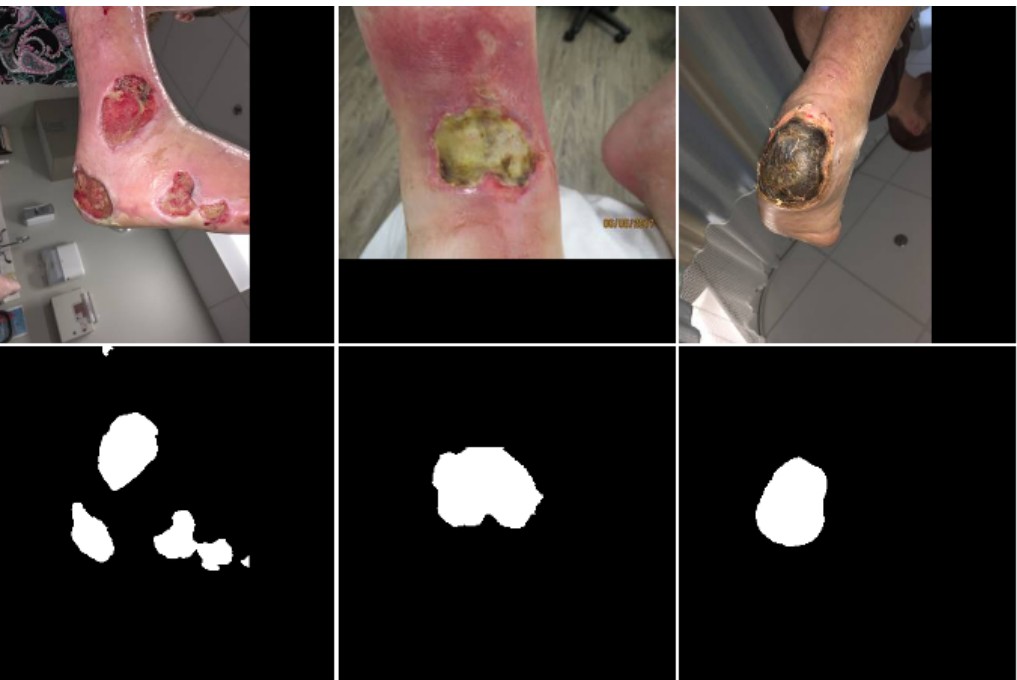

**Figure 2.** Examples of original images (**top row**) and corresponding ground-truth images (**bottom row**) from the FUSeg dataset.

### 3.3. Network Training and Testing

A standard feedforward neural network is trained as a classifier using the data generated by the procedure described in the previous subsection. The structure of the network considered was as follows: one input layer, four hidden layers with ReLU activation function, and one output layer with a sigmoid activation function. The number of neurons in the input layer depends on the length of the input vector, which in turn depends on the value of $w_{size}$. Based on the explanation provided in the previous section, the total number of neurons in the input layer is $w_{size} \times w_{size} \times 3$. The first hidden layer has the same number of neurons as the input layer, and each subsequent hidden layer has half the number of neurons as the previous layer (round down to the nearest integer where necessary). Finally, the output layer has only one neuron since only one value needs to be provided, i.e., the probability of the input vector (input sub-image) being a *wound*. The structure of the neural network model used as a classifier with $w_{size} = 9$ is provided in Figure 4.

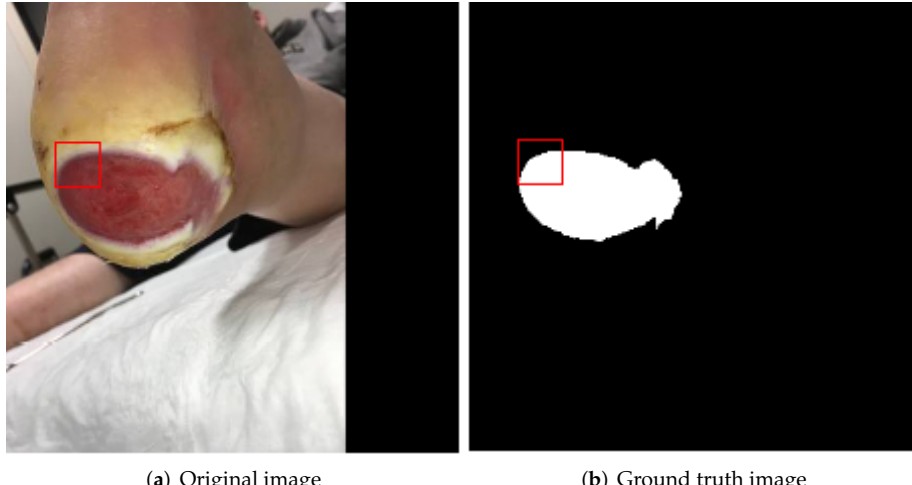

(**a**) Original image          (**b**) Ground truth image

**Figure 3.** (**a**) Input sub-image (marked by red square) defined by the current position of the sliding window on the original image (**b**) Corresponding ground-truth sub-image (also marked by red square).

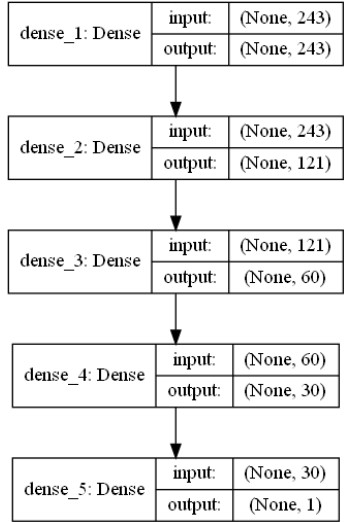

**Figure 4.** Four ReLU layers and one final layer with sigmoid function form the architecture of the proposed simple feedforward neural network. The image shows the number of neurons used in every layer for the training of the model with $9 \times 9$ input window size.

Binary cross-entropy was used as a loss function and the Adam optimizer algorithm (with corresponding default parameters defined in Keras) was used to update the network parameters. In all scenarios, 80% of the training dataset was used for model training and 20% for model validation. During model training, the precision (1), recall (2) and F1 score (3) were observed.

$$\text{Precision} = \text{TP}/(\text{TP} + \text{FP}) \tag{1}$$

$$\text{Recall} = \text{TP}/(\text{TP} + \text{FN}) \tag{2}$$

$$\text{F1} = 2 * (\text{Precision} * \text{Recall})/(\text{Precision} + \text{Recall}) \tag{3}$$

where:

TP = true positive (a pixel marked as *wound* in the predicted image and *wound* in the ground-truth image)

FP = false positive (a pixel marked as *wound* in the predicted image and *not-wound* in the ground-truth image)

FN = false negative (a pixel marked as *not-wound* in the predicted image and *wound* in the ground-truth image).

Model training was set to a maximum of 2000 epochs with a batch size of 3000 due to the large generated training data. Training of each epoch took about 68 s. Early stopping was set to monitor the validation F1 score for 200 epochs, i.e., if the score did not increase for 200 epochs, the model training was aborted. Early stopping keeps the model from overfitting and saves model training time.

The trained model is used to predict wound areas on test images. Prediction of wound areas on test images is performed as a pixel-based method. Depending on the trained model, a sliding window of appropriate window size, $w_{size}$, and $w_{step} = 1$ is used to generate test sub-images which are passed as input to the feedforward classifier. The classifier outputs a probability value between 0 and 1 based on the input sub-image. This value is assigned as the probability of the mid-pixel of the current sub-image being a wound pixel. The output image obtained after the whole input image is processed is a probability map, of the same size as the input image, indicating the probability of each pixel being a *wound* pixel. This probability map can then be multiplied by 255 to form a grayscale image. Figure 5 shows an example of the original image, the ground-truth grayscale image, and the predicted grayscale image obtained using a trained model classifier.

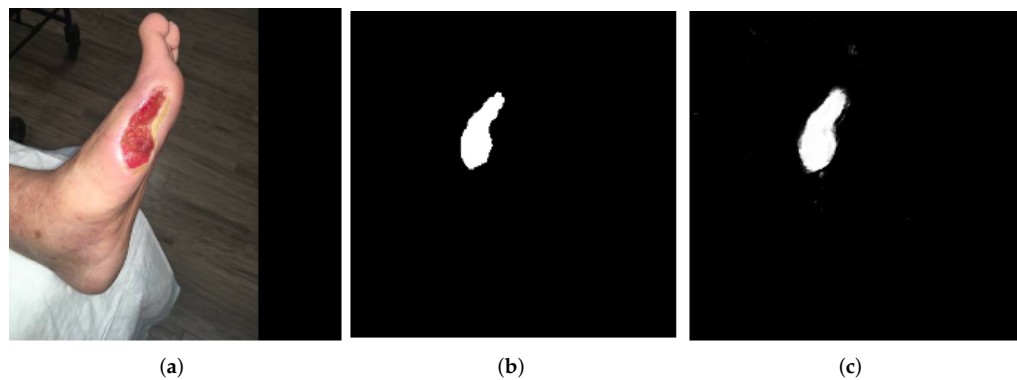

(**a**)  (**b**)  (**c**)

**Figure 5.** Prediction image of classifier obtained by training a feedforward neural network model using data generated with $w_{size} = 9$. (**a**) Original image; (**b**) Ground-truth image; (**c**) Predicted image generated by the trained model.

*3.4. Post-Processing*

For a given test image, the output of a given neural network classifier or predicted image is a probability map showing the probability of each pixel of the input image being a *wound* pixel. Post-processing is performed on the predicted images to obtain a binary image marking each pixel as either *wound* or *not-wound*. First, the predicted image or probability map is thresholded, so that the pixel values above a given threshold value are marked as *wound* and those below as *not-wound*. The value of the selected threshold influences the accuracy of the predicted binary image when compared to the given ground-truth image. This accuracy can further be improved by performing morphological operations such as opening and closing. Morphological opening first erodes the image then dilates it. It does this using the same structuring element or kernel for both operations. Morphological opening is useful for removing small objects, in this case wound pixels, while preserving the shape and size of larger objects or wounds. Morphological closing is the opposite of opening, i.e., it first dilates and then erodes the image, therefore filling small holes (filling non-wound pixels surrounded by wound pixels) while preserving the shape and size of larger objects or wounds. Thus, the post-processing of the probability map proposed in this paper has three stages that are performed sequentially:

(i)  Thresholding—a suitable threshold value ($\tau$) needs to be detected

(ii)     Morphological closing—a suitable kernel value $k_c$ (i.e., kernel dimension $k_c \times k_c$) for dilation and erosion needs to be determined

(iii)     Morphological opening—a suitable kernel value $k_o$ (i.e., kernel dimension $k_o \times k_o$) for erosion and dilation needs to be determined

Figure 6 shows an original image with the prediction result provided by the feedforward neural network, alongside the post-processed image.

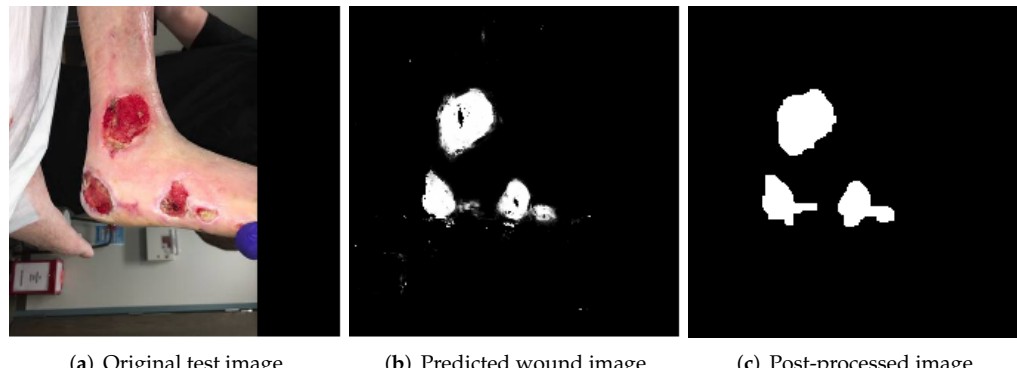

(**a**) Original test image     (**b**) Predicted wound image     (**c**) Post-processed image

**Figure 6.** An example of results obtained for the two stages of the wound-prediction process on a sample test image.

### 3.5. Ensemble of Feedforward Neural Network Classifiers

As mentioned in Section 3.3, the feedforward neural network model structure is determined by the value of $w_{size}$. In this paper, three different values of $w_{size}$ were considered, i.e., $w_{size} = 5, 7, 9$. Hence, three different classifier models were trained and tested, namely Model5, Model7 and Model9 for $w_{size} = 5, 7$ and 9, respectively. What we propose is to use an ensemble of these classifiers to obtain more accurate results. Figure 7 displays the workflow of the proposal.

An input image is processed by each of the three models separately, resulting in three post-processed images. These binary post-processed images are then combined using the AND logical operator on each corresponding pixel therefore providing a final prediction image. The rational for using the AND operator is to ensure that only pixels classified as *wound* on all binary post-processed images are taken into consideration in the final prediction image. Since these three post-processed images are obtained using different window sizes, the final result can be considered to be a combination of results obtained by performing analysis at different scales. The final binary prediction image also undergoes a post-processing procedure involving only morphological closing followed by morphological opening.

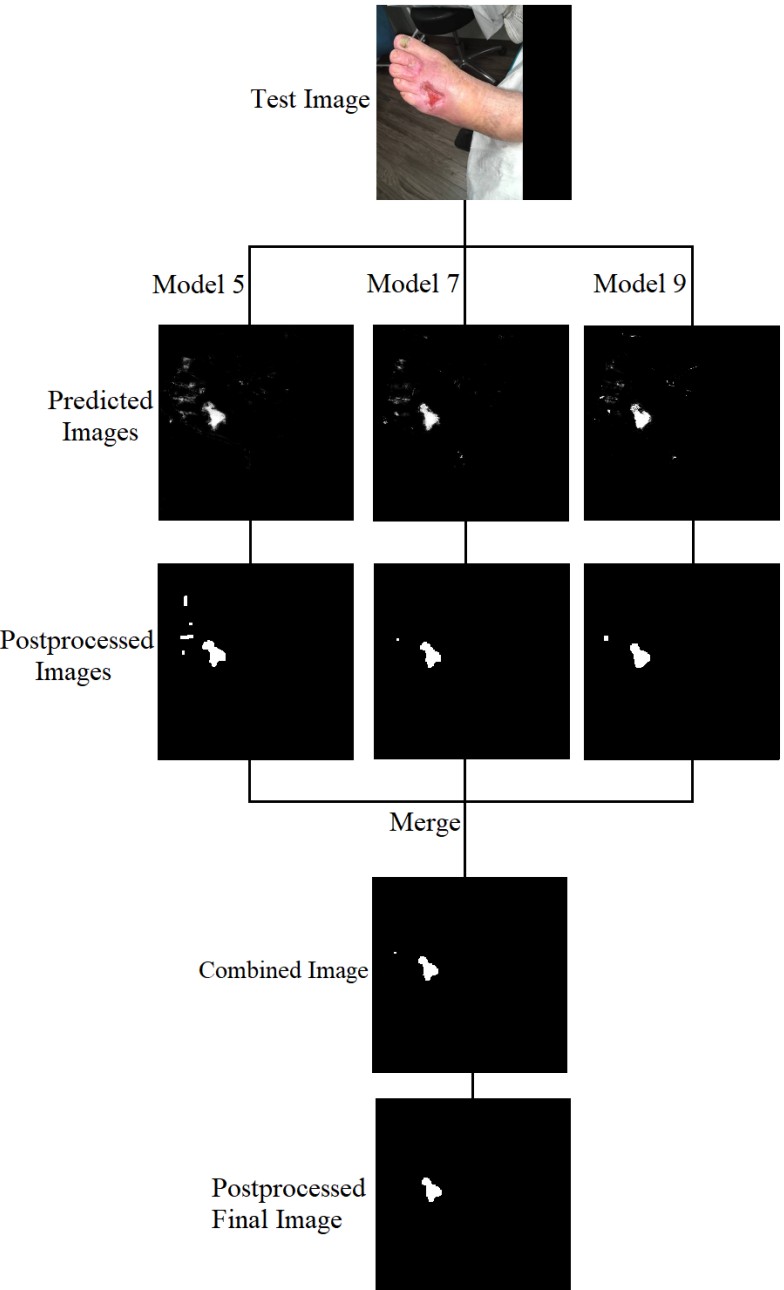

**Figure 7.** Workflow of the proposed algorithm for ensemble of feedforward neural network classifiers. Models used window sizes of $5 \times 5$, $7 \times 7$ and $9 \times 9$ respectively.

## 4. Experimental Setup

All experiments were performed on a PC with 16 GB of RAM, NVIDIA GeForce RTX 6 GB GPU, and AMD Ryzen 5 processor with six cores. All programming was done using Python programming language, specifically with Tensorflow [19] and Keras [20] libraries. Only RGB color space of the images were considered. Three different classifier models were trained and tested, namely Model5, Model7 and Model9 for $w_{size}$ = 5, 7, and 9, respectively. Training data were generated using $w_{step}$ = 1.

For the training data generated, wound_pixels$_\%$, i.e., the threshold for a given input sub-image to be marked as wound, was 50% for Model5 and Model7 while wound_pixels$_\%$ = 25% for Model9. 80% (648) of the 810 labeled training images from the MICCAI 2021 Foot Ulcer Segmentation Challenge dataset were selected at random and used in neural network

training, while 20% (162) were used for validation. Details of the feedforward neural network models and corresponding training data generated are provided in Table 1.

**Table 1.** Details of the feedforward neural network models and corresponding training data generated.

| Model Name | Wound_Pixels$_\%$ | $w_{Size}$ | Neural Network Model Structure | Number of Sub-Images Generated Per Input Image |
|---|---|---|---|---|
| Model5 | 50% | 5 | 75-75-37-18-9-1 | 48,400 |
| Model7 | 50% | 7 | 147-147-73-36-18-1 | 47,524 |
| Model9 | 25% | 9 | 243-243-121-60-30-1 | 46,656 |

The metrics: precision (1), recall (2) and F1-score (3) were analyzed at pixel level and observed during model training and evaluation. During model training, the binary cross-entropy loss function was used together with the Adam optimizer (with corresponding default parameters). Additionally, the maximum number of epochs was set to 2000 with a batch size of 3000. Early stopping was set to monitor F1-score on the validation data for 200 epochs. For a given test image, the output of each model is a probability map. To obtain a binary output image, post-processing needs to be performed. However, the post-processing stage involves the selection of suitable values of the parameters that influence the accuracy of the post-processed image compared to the ground-truth image. The three parameters that need to be determined are threshold value ($\tau$), kernel value for morphological closing ($k_c$) and kernel value for morphological opening ($k_o$). Selection of suitable parameter values was performed using the 162 validation images. For a given set of parameters, the obtained post-processed image is evaluated using the Intersection over Union (IoU) metric (4).

$$IoU = TP/(TP + FP + FN) \qquad (4)$$

Details of the procedure for determining suitable values of post-processing parameters are provided in Algorithm 1.

Analyzing Algorithm 1, it can be seen that for a given combination of ($k_c$, $k_o$), the IoU is determined for all validation images for a given threshold $\tau$, and then the average value of all IoU values obtained for all thresholds is calculated. This average IoU is used to represent the *quality* of the post-processed image for a given kernel combination ($k_c$, $k_o$). The kernel parameters that provide the maximum average IoU value are chosen as the best post-processing kernels for a specific model. Using the best processing kernel pair, the precision–recall–F1 curve is created for the given model for all thresholds. The best threshold value is then determined based on the obtained maximum F1-score.

For a given model, $k_c$ and $k_o$ were determined from the set of possible values {3,5,7,9,11,13,15} with the set of threshold values $\tau = \{0, 1, 2, \ldots, 254\}$

Figures 8–10 display the results obtained when determining the best post-processing parameters for Model5, Model7, and Model9, respectively. The obtained results are also summarized in Table 2.

**Table 2.** The best post-processing parameters obtained for the feedforward neural network classifiers.

| Model Name | Maximum Average IoU | Best Kernel Pair Values ($k_c$, $k_o$) | Maximum F1-Score | Max. Recall (for Max. F1-Score) | Threshold Value ($\tau$) (for Max. Recall) | Precision (for Selected $\tau$) |
|---|---|---|---|---|---|---|
| Model5 | 0.41 | (7,3) | 0.76 | 0.78 | 136 | 0.74 |
| Model7 | 0.48 | (5,3) | 0.81 | 0.82 | 120 | 0.8 |
| Model9 | 0.55 | (5,5) | 0.84 | 0.91 | 133 | 0.78 |

Analyzing Figure 8. for Model5, a maximum average IoU of 0.41 was obtained for $k_c = 7$ and $k_o = 3$ (Figure 8a). Using this kernel pair, and based on Figure 8b, the threshold value of $\tau = 136$ was selected, since the maximum recall value of 0.78 was obtained for the maximum F1-score of 0.76.

---

**Algorithm 1** Algorithm for determining suitable values of post-processing parameters ($k_c$, $k_o$) for a given model.

---

1: max_IOU ← 0
2: params ← ∅
3: kernel_values = {3,5,7,9,11,13,15}
4: threshold_values = {0,1,2,...,254}
5: **for** $k_c$ in kernel_values **do**
6:     **for** $k_o$ in kernel_values **do**
7:         avg_IOU ← 0
8:         **for** $\tau$ in threshold_values **do**
9:             TP ← 0
10:             FP ← 0
11:             FN ← 0
12:             **for** *each* validation_image **do**
13:                 probability_map ← Model.Predict(validation_image)
14:                 post_process1 ← Threshold(probability_map, $\tau$)
15:                 post_process2 ← MorphologicalClosing(post_process1, $k_c$)
16:                 post_process3 ← MorphologicalOpening(post_process2, $k_o$)
17:                 TP ← TP + True_Positives(post_process3, groundtruth_image)
18:                 FP ← FP + False_Positives(post_process3, groundtruth_image)
19:                 FN ← FN + False_Negatives(post_process3, groundtruth_image)
20:             IOU ← TP/(TP+FP+FN)
21:             avg_IOU ← avg_IOU + IOU
22:         avg_IOU ← avg_IOU/255
23:         **if** avg_IOU > max_IOU **then**
24:             max_IOU ← avg_IOU
25:             params ← {$k_c$,$k_o$}

---

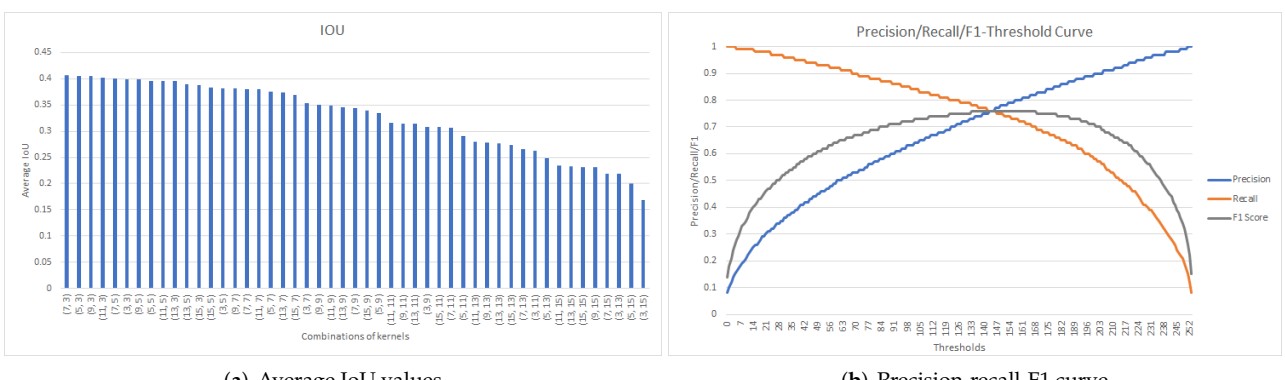

(**a**) Average IoU values            (**b**) Precision-recall-F1 curve

**Figure 8.** Results obtained for Model5 (**a**) Average IoU values for all kernel combinations and (**b**) Precision-Recall-F1 curve for the best kernel combination.

Performing a similar analysis for Model7 (Figure 9), a maximum average IoU of 0.48 was obtained for $k_c$ = 5 and $k_o$ = 3 (Figure 9a). Using this kernel pair, and based on Figure 9b, the threshold value of $\tau$ = 120 was selected, since the maximum recall value of 0.82 was obtained for the maximum F1-score of 0.81.

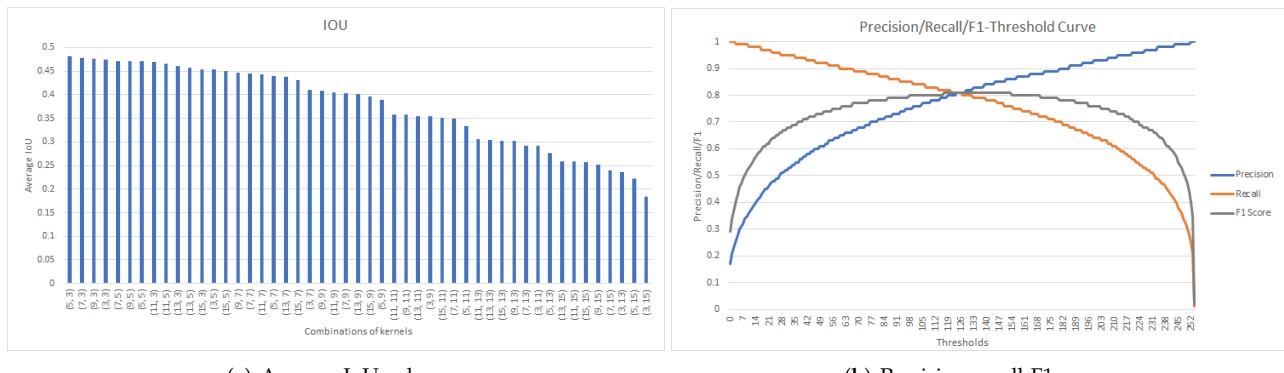

(**a**) Average IoU values     (**b**) Precision-recall-F1 curve

**Figure 9.** Results obtained for Model7 (**a**) Average IoU values for all kernel combinations and (**b**) Precision-Recall-F1 curve for the best kernel combination.

The best kernel pair for Model9 was obtained for $k_c$ = 5 and $k_o$ = 5 (Figure 10a) with corresponding maximum average IoU of 0.55. Using this kernel pair, and based on Figure 10b, the threshold value of $\tau$ = 133 was selected, since the maximum recall value of 0.91 was obtained for the maximum F1-score of 0.84.

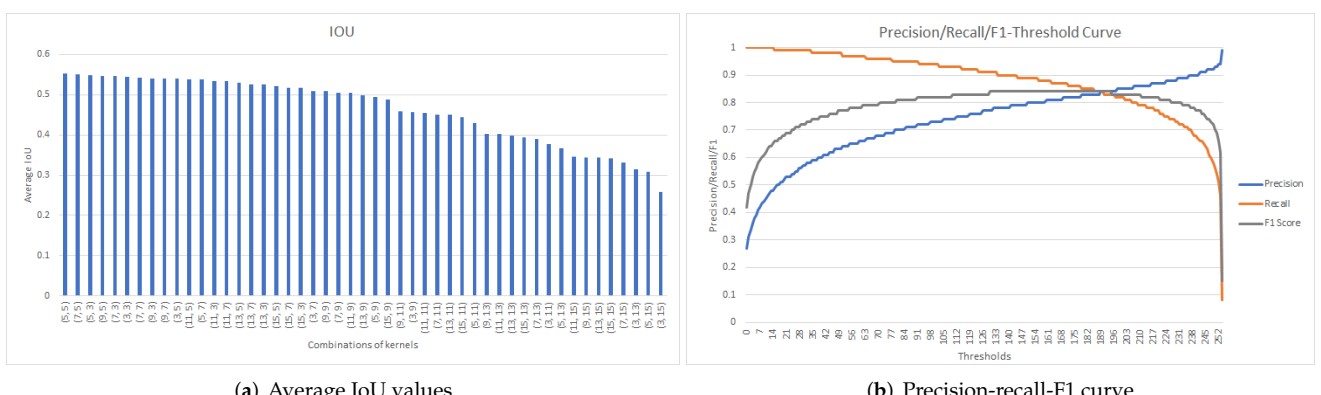

(**a**) Average IoU values     (**b**) Precision-recall-F1 curve

**Figure 10.** Results obtained for Model9 (**a**) Average IoU values for all kernel combinations and (**b**) Precision-Recall-F1 curve for the best kernel combination.

After the three models were completely defined, the ensemble of feedforward network classifier approach was analyzed. Two slightly different ensemble classifiers were implemented. The first ensemble model, referred to herein as $Model_{En\_1}$, involves using the previously three defined models (with corresponding best kernel values and threshold value) to generate three binary post-processed images for the same input image. These binary post-processed images are then combined using the AND logical operator to create a final binary prediction image which also undergoes a final post-processing step (see Section 3.5 and Figure 7). Thus, an additional kernel pair ($k_c$, $k_o$) needs to be determined. For a given kernel pair, the average IoU was determined on all validation images. The best kernel pair values were selected based on the maximum average IoU. The results of this experiment are displayed in Figure 11. The best kernel pair for the final post-processing procedure for $Model_{En\_1}$ was $k_c$ = 15 and $k_o$ = 3 with a maximum average IoU value of 0.60.

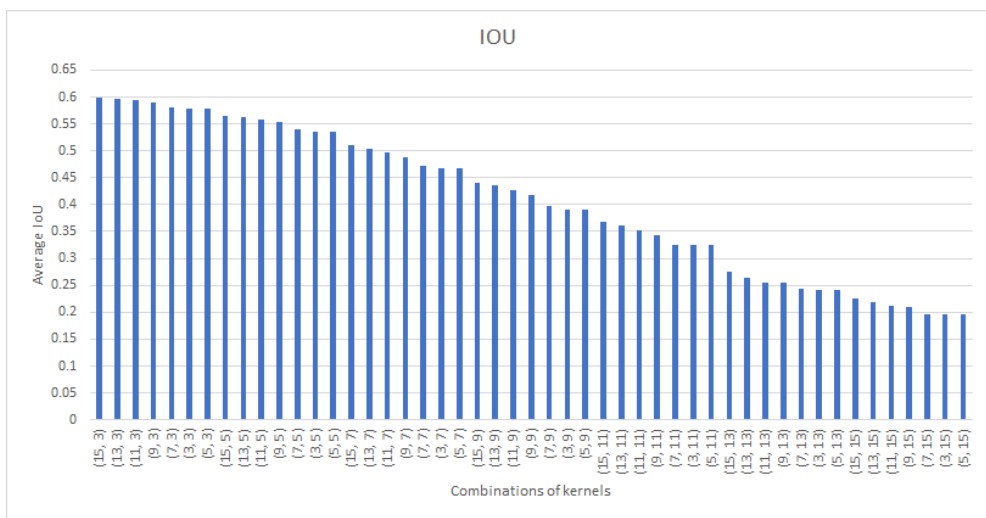

**Figure 11.** Average IoU values. Results obtained for Model$_{En\_1}$ average IoU values for all kernel combinations.

The second ensemble model, referred to herein as Model$_{En\_2}$, involves using the three previously defined models (with corresponding best kernel values and without their previously determined threshold values). For this model, the additional kernel pair ($k_c$, $k_o$) needs to be determined for the final post-processing step, as well a threshold that is the same for all the three previously defined models. This was done similarly to the previously described procedures. For a given kernel pair, the average IoU was determined for all validation images and for all threshold values. The best kernel pair values were selected based on the maximum average IoU. The results of this experiment are displayed in Figure 12. The best kernel pair for Model$_{En\_2}$ was $k_c$ = 15 and $k_o$ = 3 with a maximum average IoU value of 0.49 (Figure 12a). Using this kernel pair, and based on Figure 12b, the threshold value of $\tau$ = 82 was selected, since the maximum recall value of 0.87 was obtained for the maximum F1-score of 0.84.

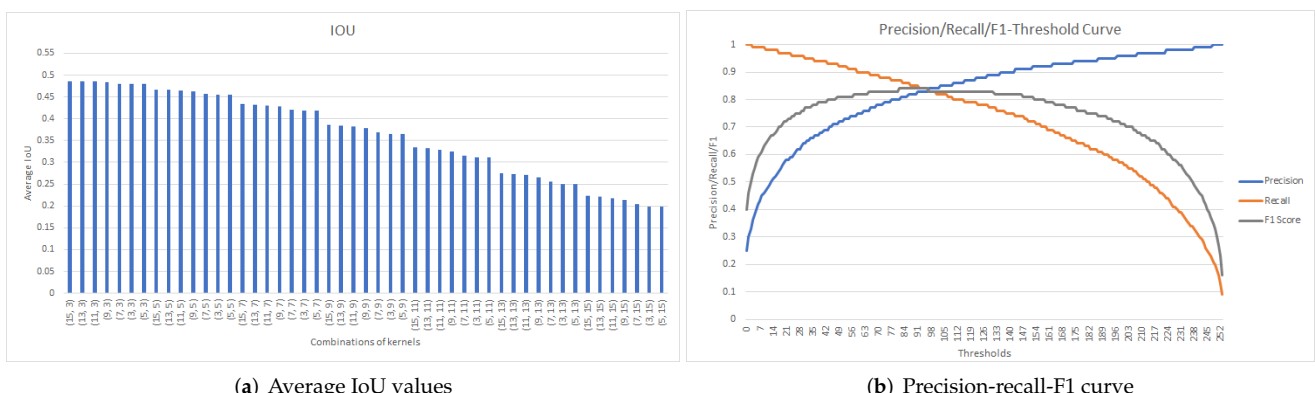

(**a**) Average IoU values                    (**b**) Precision-recall-F1 curve

**Figure 12.** Results obtained for Model$_{En\_2}$ (**a**) Average IoU values for all kernel combinations and (**b**) Precision-Recall-F1 curve for the best kernel combination.

An overview of the ensemble classifiers is provided in Table 3.

**Table 3.** The best post-processing parameters obtained for the feedforward neural network classifiers.

| Ensemble Model Name | Parameters | Neural Network Classifiers Used | | | Maximum Average IoU | Best Kernel Pair Values ($k_c$, $k_o$) (Final Post-Processing) | F1-Score | Recall | Precision |
|---|---|---|---|---|---|---|---|---|---|
| | | Model5 | Model7 | Model9 | | | | | |
| $Model_{En\_1}$ | Kernel pair ($k_c$, $k_o$) | (7,3) | (5,3) | (5,5) | 0.60 | (15,3) | 0.83 | 0.77 | 0.89 |
| | Threshold ($\tau$) | 136 | 120 | 133 | | | | | |
| $Model_{En\_2}$ | Kernel pair ($k_c$, $k_o$) | (7,3) | (5,3) | (5,5) | 0.49 | (15,3) | 0.84 | 0.87 | 0.81 |
| | Threshold ($\tau$) | | 82 | | | | | | |

As pointed out in the Introduction, the wound-detection method proposed in this paper is the first phase of a more complex system being developed, and it is imperative that all wounds are detected. Hence the number of false negative pixels should be minimum, implying the recall should be maximum. Thus, the main criterion for selecting the most suitable classifier is based on the recall value on the validation set. Analyzing the data in Tables 2 and 3, Model9 (recall = 0.91) is selected as the most suitable single neural network classifier, with $Model_{En\_2}$ (recall = 0.87) being the most suitable ensemble classifier.

All five models were evaluated and compared using the test dataset. The results obtained are given in Table 4.

**Table 4.** Classification results on the test dataset.

| Model Name | F1-Score | Precision | Recall | IoU |
|---|---|---|---|---|
| Model5 | 0.69 | 0.67 | 0.70 | 0.52 |
| Model7 | 0.71 | 0.69 | 0.72 | 0.55 |
| Model9 | 0.71 | 0.68 | 0.74 | 0.55 |
| $Model_{En\_1}$ | 0.73 | 0.80 | 0.67 | 0.57 |
| $Model_{En\_2}$ | 0.74 | 0.72 | 0.77 | 0.59 |

It can be noticed in Table 4 that Model9 is the best single neural network classifier, since the highest values among the single neural network classifiers were obtained for F1-score (0.71), recall (0.74) and IoU (0.55). $Model_{En\_2}$ classifier is the best ensemble classifier as well as the overall best classifier, since the highest values were obtained for F1-score (0.74), recall (0.77) and IoU (0.59).

$Model_{En\_2}$ was compared to the segmentation model proposed in [15]. The authors in [15], Wang et al., compared their proposed model to four other models, namely VGG16, SegNet, U-Net and Mask-RCNN, using the Medetec dataset and showed their model to be superior. The method proposed by [15] was implemented using the code provided by the authors. Their deep neural network was trained using the settings (threshold = 130) and post-processing method described in the [15]. Training was performed using the same training dataset used in this paper. The results obtained are displayed in Table 5, together with those obtained by $Model_{En\_2}$.

**Table 5.** Comparison of Wang et al. [15] model and proposed ensemble model using test dataset (200 images) from FuSeg dataset.

| | F1-Score | Precision | Recall | Total Time for 200 Test Images (Prediction + Post-Processing) |
|---|---|---|---|---|
| Wang et. al. [15] | 0.79 | 0.85 | 0.74 | 30.1 (3.5 + 26.6) s |
| Model$_{En\_2}$ | 0.74 | 0.72 | 0.77 | 69.9 (69.8 + 0.1) s |

It can be noticed that even though the model proposed in [15] has a higher precision and F1-score, the ensemble classifier proposed in this paper has a higher recall value. Additionally, even though the total classification time of the ensemble model is about 2.3 times that proposed by [15], it is still fast enough (0.35 s per image) to be used in a wound-detection system.

To test the robustness of the two models provided in Table 5 on image rotation, four new test sets using the original 200 test images from the FuSeg dataset were generated:

(a)     All original test images were flipped horizontally.
(b)     All original test images were flipped vertically.
(c)     All original test images were rotated by −45°.
(d)     All original test images were rotated by +45°.

The two models are tested on these test sets, and the results are provided in Table 6.

**Table 6.** Comparison of model robustness to image rotation.

| Test Set | Model | F1-Score | Precision | Recall | Percentage Change Compared to Original Test Set (Table 5) | | |
|---|---|---|---|---|---|---|---|
| | | | | | F1-Score | Precision | Recall |
| Original test images flipped horizontally | Wang et al. [15] | 0.71 | 0.82 | 0.62 | −10.1 | −3.5 | −16.2 |
| | Model$_{En\_2}$ | 0.71 | 0.68 | 0.75 | −4.1 | −5.6 | −2.6 |
| Original test images flipped vertically | Wang et al. [15] | 0.75 | 0.81 | 0.69 | −5.1 | −4.7 | −6.8 |
| | Model$_{En\_2}$ | 0.73 | 0.70 | 0.77 | −1.4 | −2.8 | 0.0 |
| Original test images rotated by −45° | Wang et al. [15] | 0.61 | 0.82 | 0.49 | −22.8 | −3.5 | −33.8 |
| | Model$_{En\_2}$ | 0.69 | 0.75 | 0.64 | −6.8 | +4.2 | −16.9 |
| Original test images rotated by +45° | Wang et al. [15] | 0.60 | 0.84 | 0.47 | −24.1 | −1.2 | −36.5 |
| | Model$_{En\_2}$ | 0.69 | 0.75 | 0.64 | −6.8 | +4.2 | −16.9 |

When comparing the F1-score and recall values of the proposed ensemble model obtained on the new test sets to those obtained on the original test set, there is a slight percentage decrease in values for test images flipped either vertically or horizontally. The percentage decrease is a bit greater for test images rotated by either +45° or −45°. On the other hand, the percentage decrease in values for the model proposed by Wang et al. [15] is much greater compared to the proposed ensemble model.

The ensemble classifier proposed in this paper is used as a wound detector. Thus, all predicted or detected wound clusters are marked using bounding boxes. Figure 13 shows several images with detected wounds marked with bounding boxes. The predicted wound is marked with a blue bounding box. If there are several predicted wounds on the same image, all the predicted wounds are then marked with red bounding boxes, while the largest predicted wound is marked with a blue bounding box. Ground-truth wounds are marked using green bounding boxes.

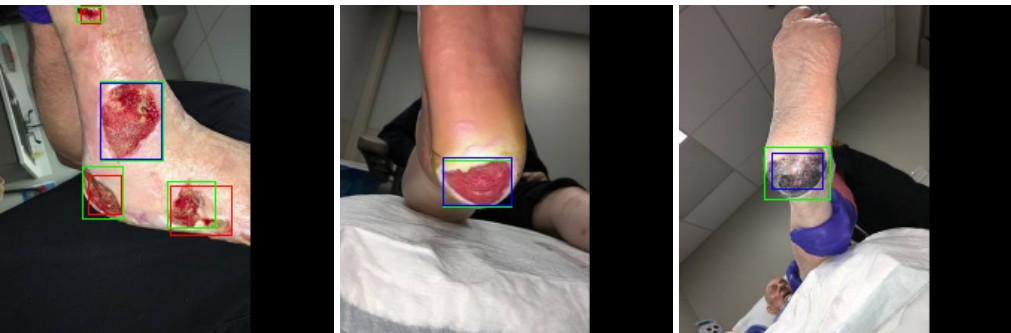

**Figure 13.** Detected wound areas marked on test images with bounding boxes. Blue color represents the largest detected wound, red bounding boxes mark other wounds detected on the image while green bounding boxes are generated from ground-truth image.

## 5. Discussion

Analyzing the feedforward neural network classifiers (Table 2), increasingly better results in terms of F1-score, IoU, recall and precision are obtained with increasing values of $w_{size}$. This can be attributed to the fact that the larger the window size, the more information is provided to the classifier, therefore predicting wound pixels with a greater accuracy. Using an ensemble of classifiers also increases the accuracy of the predictions, especially if the AND logical operator is used in combining the predictions of all the sub-classifiers of the ensemble. The proposed ensemble model is also much more robust to image rotations compared to the model proposed by Wang et al. [15]. The network structure of convolutional neural networks is such that they mainly take into account the spatial structure of data, while simple feedforward neural networks rely more on the information content of data. Since wounds are highly irregular and can be of any shape and/or texture, this robustness might be explained by the fact spatial structure of data is not as important as the information content, implying that the sub-images created by the sliding window contain enough information for classification to be performed by simple feedforward networks. However, further experiments need to be performed for this statement to be corroborated.

## 6. Conclusions

The wound-detection method proposed in this paper is based on a simple five-layered feedforward neural network. A standard fixed-size overlapping sliding-window procedure is used to generate input data for the neural network classifier. Three different sliding-window sizes were considered. As a result, three different neural network classifiers were created and trained. The predicted outputs of the neural network classifiers were further processed using three steps: thresholding, morphological closing and morphological opening. This was performed to increase the accuracy of the predicted images. The proposed image post-processing procedure was shown to be robust and fast. Better prediction capabilities were noticed for neural network models created and trained with data generated by bigger sliding-window sizes. This can be explained using the fact that the larger the window size, the more information is provided to the classifier, therefore predicting wound pixels with a greater accuracy. An ensemble of neural network classifiers formed using the trained three feedforward neural network classifiers is also proposed in this paper. The binary post-processed images obtained as output predictions of the three neural networks are combined using the AND logical operator. Better prediction results were obtained using ensemble classifiers compared to the three single feedforward neural network classifiers. The ensemble classifier proposed in this paper gives satisfactory results in terms of recall and processing time, is relatively robust to image rotations, and is implemented as a wound detector as part of a larger system for wound analysis. Even though wounds are highly irregular and can be of any shape or texture, the results obtained

in this paper indicate that simple feedforward neural networks are suitable for wound detection and there is no need for more complicated neural networks. Future research will focus on detecting the right neural network structures, i.e., the selection of the optimal number of layers and neurons, as well as suitable activation functions, in order to improve upon the current results obtained. Possible preprocessing steps, such as conversion to other color spaces and color correction, will also be considered. The effect of data augmentation of the training set on classification performance, especially with respect to image rotation and flipping, will also be analyzed.

**Author Contributions:** Conceptualization, E.K.N.; methodology, D.M. and E.K.N.; software, D.M. and E.K.N.; validation, D.M. and E.K.N.; writing—original draft preparation, D.M.; writing—review and editing, E.K.N. and D.F.; supervision, D.F. All authors have read and agreed to the published version of the manuscript.

**Funding:** The paper is developed under project "Methods for 3D reconstruction and analysis of chronic wounds" and is funded by the Croatian Science Foundation under number UIP-2019-04-4889.

**Institutional Review Board Statement:** Approval from the ethical board of Clinical Medical Center Osijek, Croatia, no. R2-4787/2019. has been obtained before any patient recording has been made.

**Informed Consent Statement:** Informed consent was obtained in written form, from all patients involved in the study.

**Conflicts of Interest:** The authors declare no conflict of interest.

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
