# Peer review of "Wound Detection by Simple Feedforward Neural Network"

_electronics, doi:10.3390/electronics11030329_

Round 1

Reviewer 1 Report

The paper presents research for detecting the wound area using the feedforward neural network. The proposed method can be used to segment and define the wound area. The wound objects are collected from different angles and lighting environments. Some comments need to be addressed by the authors:

1) What are the morphological (opening and closing) settings used in the paper? What is the filter dimension used in the morphological implementation? 

2) Does the algorithm need to apply morphological filtering for an image more than once? Is there any different setting when the wound appears in different colours, such as reddish or yellowish for the other cases? 

3) In some cases, the wound surface is not perpendicular to the lens surface, as shown in Figure 13 (the centre and right images). The wound surface can be perpendicular to the lens surface, as depicted in Figure 13 (the left image).  This different capturing angle can deviate the image resolution. How does the algorithm deal with a spatial resolution divergency in the image? 

4) The proposed algorithm should be tested with the rotated image at different rotation angles. The test aims to study the algorithm robustness with the rotation adjustment. 

5) The authors need to add more comparable studies in Table 5. The authors should cover another segmentation method based on deep learning in the table. Implementation of U-Net architecture could be considered by the authors.

Reviewer 2 Report

This work presents 'Wound detection by simple Feedforward Neural Network'. Three models were trained and tested using some images.  The proposed idea is simple and I have a few suggestions. 

  1. A few important parameters related to the NN is missing. Please discuss.
  2. This work is not compared with any other scheme/model. 
  3. Discuss all accuracy metrics in the abstract. 
  4. In conclusion, what do you mean by 'suitable preprocessing steps'?
  5. You have discussed a number of accuracy metrics. Could you please outline any other metrics (if available)? 
  6. Why Relu is used? Why not any other activation function.
  7. Is it possible to train the model using a number of hidden layers (not 4 layers). Why 4 is used in this work? 

Reviewer 3 Report

This work presents a standard feedforward neural network as a simple yet useful tool for the purpose of wound segmentation, which is part of a compact system that analyses wounds. This work is a good demo of how AI can contribute to medical treatment and nursing. The research is well designed and the results are convincing, while the quality of the manuscript can be further improved with better-organized figures. 

Round 2

Reviewer 1 Report

Dear Authors,

All reviewer comments have been addressed by the author.

Best regards

Reviewer 2 Report

The Paper is revised based on my comments. Accepted in present form.